# The Glass-Transition Temperature of Supported PMMA Thin Films with Hydrogen Bond/Plasmonic Interface

**DOI:** 10.3390/polym11040601

**Published:** 2019-04-02

**Authors:** Jiayao Chen, Jing Li, Lirong Xu, Wei Hong, Yuzhao Yang, Xudong Chen

**Affiliations:** 1Key Laboratory for Polymeric Composite and Functional Materials of Ministry of Education of China, Guangdong Engineering Technology Research Center for High-performance Organic and Polymer Photoelectric Functional Films, School of Chemistry, Sun Yat-sen University, Guangzhou 510275, China; chenjy248@mail2.sysu.edu.cn (J.C.); lijing275@mail.sysu.edu.cn (J.L.); xulr@mail2.sysu.edu.cn (L.X.); hongwei9@mail.sysu.edu.cn (W.H.); 2Institute of New Energy Technology, College of Information Science and Technology, Jinan University, Guangzhou 510632, China

**Keywords:** glass-transition temperature, temperature dependent fluorescence, surface plasmon, interfacial effect, conformation

## Abstract

The interfacial effect is one of the significant factors in the glass-transition temperature (T_g_) of the polymeric thin film system, competing against the free surface effect. Herein, the T_g_s of poly (methyl methacrylate) (PMMA) films with different thicknesses and substrates are studied by fluorescence measurements, focusing on the influence of interfacial effects on the T_g_s. The strong interaction between PMMA and quartz substrate leads to increased T_g_s with the decreased thickness of the film. The plasmonic silver substrate causes enhanced fluorescence intensity near the interface, resulting in the delayed reduction of the T_g_s with the increasing film thickness. Moreover, as a proof of the interface-dependent T_g_s, hydrogen bonds of PMMA/quartz and molecules orientation of PMMA/silver are explored by the Raman spectroscopy, and the interfacial interaction energy is calculated by the molecular dynamics simulation. In this study, we probe the inter-relationship between the interfacial interactions arising from the different substrates and the T_g_ behavior of polymer thin films.

## 1. Introduction

The glass-transition temperature (T_g_) is considered a significant physical property in amorphous polymer. In recent years, the study of T_g_s of polymer thin films has drawn much attention [1,2]. As the thickness of polymer thin films decreases, T_g_ of polymer thin film will deflect from the bulk film because of the conformation changes in macromolecular chains, aggregation structure, and dynamical property [3,4]. Numerous studies have explored the thickness dependence of T_g_ in free-standing polymer thin films or supported polymer thin films via Differential Scanning Calorimeter (DSC), Reflection–Absorption Fourier Transform Infrared Spectroscopy (FT-IR), ellipsometry, Brillouin Light Scattering (BLS), X-ray reflectivity measurement, fluorescence spectroscopy [5,6,7,8,9,10,11,12,13,14,15,16]. The T_g_ alteration can be explained by the effects of free surface, chain confinement, interfacial interaction, molecule weight, and finite-size [17,18,19,20,21].

The measurement of T_g_s by temperature dependence fluorescence has drawn considerable attention in decades, including fluorescence based on the polymer chains and extrinsic probes. Brady and Charlesworth first reported that the intrinsic fluorescence of an epoxy resin could be used for T_g_ measurement with good sensitivity [22]. Torkelson’s group studied the T_g_ measurement of extrinsic fluorescence with multiple rotor probes and chromophore labeled polymer [6,23]. In 2002, they first reported on the fluorescence measurement of chromophore labeled polymers thin films for determining the effects of film thickness on T_g_, and enhancing the sensitivity through the time-dependence fluorescence intensity [6]. The determination of the T_g_ distribution within polymer thin film via fluorescence measurements was further refined, allowing the characterization of T_g_s of the surface, interior and interface [7]. In addition to the single component films, T_g_ characterization by fluorescence for multilayer films and diblock copolymers ultrathin films were developed [24].

Instrumental analysis has revealed the structural, morphological, and spectroscopic details of the behavior of polymer thin films over a broad range of length scales [25]. However, a limited number of studies have unveiled the influence of interfacial effects of substrates on T_g_ values from the segmental and bond levels. Besides, the interfacial effects on the fluorescence are complex, especially for the metal surface which could have fluorescence enhancement through surface plasmon effects. To understand how the superficial/interfacial effect induces the orientation to the polymer chain at an interface by specific interactions, how the molecule mobility delivers from surface/interface to the bulk and what kind of interaction dominates the interfacial behavior, adequate methods should be conducted to study the interfacial effects on T_g_ of polymer thin film, with and without a plasmonic substrate.

Herein, the fluorescence spectroscopy managed to measure the T_g_ of fluorescent probe-labeled PMMA films with distinct thickness on substrates with robust interaction and plasmonic effects for examining the interfacial effects on T_g_. Moreover, Raman spectroscopy was used to explore the interaction between PMMA molecules and substrates. Molecular dynamics (MD) simulations were executed to calculate the interfacial interaction energy, further investigating the binding mechanisms of PMMA on the quartz/silver surface. We supposed that this work would be a significant contribution to the understanding of plasmonic effects on T_g_ of ultrathin film from the perspective of intermolecular interaction.

## 2. Materials and Methods 

### 2.1. Materials

The 4-tricyanovinyl-[N-(2-hydroxyethyl)-N-ethyl]aniline (TC1)-labeled PMMA was synthesized according to Torkelson et al. [26] as follows (Appendix A): (1) Tetracyanoethylene (4.5 g) and 2-(*N*-ethylanilino) ethanol (6.612 g) were reacted in dimethyl formamide (23 mL) at 55 °C for 15 min. After recrystallized with acetic acid in the ice and water bath, the TC1was obtained. (2) TC1-labeled methacrylate monomer was synthesized by the esterification reaction of methacryloyl chloride (0.4 mL) and TC1 (0.838 g) in the presence of dichloromethane (0.4 mL) and triethylamine (0.4 g) stirring at 0 °C for 2 h. The reacted solution was washed with sodium bicarbonate and disposed to roto-evaporation for removing the solvent. The obtained product was recrystallized in ethanol/water solution. (3) TC1-labeled PMMA (*M*_w_ = 260,000 g/mol, PDI = 1.72 by Gel Permeation Chromatography, GPC) was synthesized through free radical polymerization by TC1-labeled (0.1 g) and unlabeled (10 mL) methacrylate monomer with initiator azodiisobutyronitrile (0.005 g). The synthesis of the compound and polymer mentioned above would be confirmed by FTIR and ^1^H Nuclear Magnetic Resonance (NMR), and the content of TC1 probe in PMMA was determined by UV/visible absorbance spectroscopy. In the Raman characterization, pure PMMA without probe molecules would be used and it was synthesized in the same way. Chemicals were purchased from Sigma-Aldrich (Aldrich Chemical Co., Milwaukee, WI, USA) and Aladdin Reagent (Shanghai, China). All solvents were purified by distillation before used.

### 2.2. Sample Preparation

Films were prepared by spin-casting from filtered toluene (chromatographically pure, Aladdin Reagent, Shanghai, China) solutions onto cleaned quartz and silver substrates. The thicknesses ranging from 25–1000 nm were varied by changing the solution concentration and spin-cast speed. The samples of coating PMMA bulk layer were spin-casted onto KBr infrared crystal windows (Botianshengda Co. Ltd, Beijing, China) [7]. After being dried and annealed in the vacuum oven at T_g_ + 20 K for 20 min, the KBr substrate coated PMMA was removed to the beaker with deionized water. As KBr was dissolving, the PMMA film floated on water and transferred onto the required sample. All prepared samples were dried and annealed in a vacuum at T_g_ + 20 K for 20 min. The bulk TC1-labeled PMMA without any substrates was prepared by a thermo-compressor (at T_g_ + 20 K) which is an accessory of FT–IR Analyzer, and annealed in a vacuum at T_g_ + 20 K for 20 min to eliminate stress.

### 2.3. Sample Measurement and Data Analysis

The T_g_ of PMMA-TC1 thin film was measured using a FLS980 Spectrometer (Edinburgh Instruments, Livingston, UK) with an OptistatCF Liquid Helium Cryostat (Oxford Instruments, Oxfordshire, UK), which could be obtained by the intersection of linear fits to the rubbery and glassy state of the fluorescence emission intensity in the temperature dependence process. The excitation wavelength was 510 nm, and emission spectra were collected from 520 to 820 nm. In order to filter the light source signal, the fluorescence filter (520 nm) was used for all samples. After having annealed at 443 K for 20 min, samples were recorded by an emission intensity measurement at 5 K increments from 353 K to 463 K allowing 5 min for equilibration at each temperature setting. The T_g_ can be obtained by the intersection temperature of the linear fits in two segments of fluorescence emission integrated intensity, and typical correlations (R^2^) are better than 0.990 at all emission wavelengths. To avoid various peaks of the fluorescence intensity data in the emission spectra, integrated intensities were managed to identify T_g_ [27].

To characterize the intermolecular interaction of samples above, a Laser Micro-Raman Spectrometer (Renishaw inVia-Reflex, New Mills, UK) was used to record Raman spectra equipped with an Ar^+^ laser as the excitation resource (532 nm). Spectra were typically obtained within 50% of laser power, 30 s of exposure time, 3 times of accumulation and 1 cm^−1^ of resolution.

The silver substrates were obtained by the evaporated deposition of 80 nm Ag on top of the quartz with the base pressure of 1 × 10^−5^ Pa (vacuum deposition equipment, Sky technology development, Shenyang, China). The FTIR spectra were collected by FT-IR Analyzer (Nexus 670, Thermo Nicolet, Madison, WI, USA). The ^1^H NMR spectra of TC1 and TC1-labeled monomer were recorded on spectrometer operating at 300 MHz frequency (^1^H-NMR, Bruker Avance, Bremen, Germany) (CDCl_3_, tetramethylsilane (TMS) as the internal standard). The content of TC1-label was determined by UV/visible absorbance spectroscopy (UV-3600 Shimadzu Co., Kyoto, Japan). GPC analysis was measured by Waters Alliance GPC2000 system using Styragel linear columns in tetrahydrofuran (THF) at 40 °C. The T_g_ of bulk PMMA was characterized by Differential Scanning Calorimeter (DSC-4000, Perkin Elmer, Akron, OH, USA) at heating rate of 10 K/min on second heating. The PMMA film thickness was determined by step profile (ET 150, Kosaka Lab, Tokyo, Japan).

### 2.4. Molecule Models and Simulation Details

In the MD simulation, Materials Studio^®^ in the COMPASS force field was used. The PMMA was modeled as an amorphous thin film structure consisting of 72 isotactic chains, and each PMMA chain had 10 unit monomers. All simulations were performed in cuboid simulation boxes and periodic boundary conditions were implemented in all three dimensions. The simulation run was carried out by geometry optimization, and a stable aggregation structure of PMMA with initial density at 1 g/cm^3^ was obtained. The silver and quartz substrates were modeled as flat plates, and a PMMA thin film was placed on above the substrates on one side and exposed to vacuum on the other side. The (0 0 1) crystallographic face of quartz and (1 1 1) crystallographic face of silver were used to model the solid substrate. To obtain a chemically realistic surface, all the non-bridging oxygen atoms of quartz were protonated. Since substrate materials of silver and quartz were much harder and stiffer than the PMMA, all the atoms in the substrate were constrained. The simulation run was performed for a canonical number of atoms, volume and temperature (NVT) ensemble for 500 ps at 300 K. Since adhesion between PMMA and substrate was primarily dependent on the intrinsic non-bond interaction characteristics, the interfacial interaction energy (adhesion energy) between PMMA molecules and substrate could be calculated as:E_int_ = E_total_ − E_PMMA_ − E_subs_(1)
where E_total_ was the total potential energy of the whole system containing all atoms, E_PMMA_ was the local potential energy of PMMA, and E_subs_ was the local potential energy of substrate.

## 3. Results and Discussion

### 3.1. Fluorescence of TC1-Labeled PMMA Films on Quartz and Silver Substrates

In order to investigate the PMMA thin film with different interface and surface effects, we performed four structures as follows (Figure 1): TC1-labeled PMMA thin film supported on the quartz substrate was denoted as “structure-A”; TC1-labeled PMMA thin film coated with 200 nm bulk PMMA film supported on the quartz substrate was denoted as “structure-B”; TC1-labeled PMMA thin film supported on silver substrate (80 nm by evaporated deposition) was denoted as “structure-C”; TC1-labeled PMMA thin film coated with 200 nm bulk PMMA film supported on the silver substrate (80 nm by evaporated deposition) was denoted as “structure-D”, respectively. The optical images of the four TC1-labeled PMMA samples were provided in Appendix A.

T_g_s of PMMA thin films were evaluated by the fluorescence measurement with the TC1 probe that was covalently labeled to PMMA chain at trace levels. The synthesis of the TC1, TC1-labeled MMA monomer and TC1-labeled PMMA were all confirmed by NMR and FT-IR in Appendix A. The TC1-labeled PMMA (*M*_n_ ≈ 151,000 g/mol, *M*_w_ ≈ 260,000 g/mol, PDI = 1.72 by GPC) contained 1 wt% TC1-labeled monomer, determined by UV/visible absorbance spectroscopy via the absorption-concentration standard curve of TC1-labeled MMA (Figure 2). Steady-fluorescence was used to study the plasmonic effects on the fluorescence with structure-A (without plasmonic effect) and structure-C (with plasmonic effect) at room temperature. As shown in Figure 2b,c, the fluorescence intensity of samples on silver substrates was much larger than that on quartz substrates for both the ultrathin film and the bulk film, which illustrates the enhanced fluorescence by surface plasmon effects. The peak positions of the samples with silver substrates red-shifted (from 578 nm to 606 nm) with increasing film thickness, while the peak positions of samples with quartz exhibited no change (at 595 nm). The repetitive experiments of fluorescence emission spectra were carried out and summarized in Appendix A. The fluorescence probe TC1 adjacent to metal was affected by the strong electromagnetic field, which might lead to the enhancement of the excitation rate of luminescent molecule significantly [28]. The luminescence enhancement ratios of samples on silver substrate were 9.25, 9.55, 6.21 and 5.79 for film thickness of 25, 110, 230 and 760 nm, respectively, as compared with the samples on quartz substrates.

### 3.2. Effects of Thickness and Substrate on the T_g_s of PMMA Films

The steady fluorescence measurements were performed to monitor the annealed PMMA-TC1 thin films ranging from 353–463 K. By plotting the integrated fluorescence intensity as a function of temperature, the inflection point of the curve could be regarded as the value of T_g_ according to earlier fluorescence studies [6,7,8,27]. DSC was used to measure the T_g_ of bulk PMMA-TC1 to verify the reliability of fluorescence measurement for T_g_ characterization. The mid-point of the specific step in DSC was defined as T_g_, so the T_g_s of PMMA bulk powder was 399.12 K. As is well known, there is a process in glass transition of amorphous polymer between glass state and flexible state. At the beginning of inflection point arising in DSC curves (see in Appendix A), some segments had a certain freedom of movement, while others did not within the long chain of PMMA, which defined as T_g_ of onset (390.46 K). When all of the long chain molecules mobilized, the point of temperature considered to be T_g_ of termination (403.65 K). The T_g_ of bulk TC1-labeled PMMA film (≈1000 nm) supported on quartz substrates obtained by the fluorescence method was about 416.25 K (Figure 3a), indicating that the measured T_g_ in this work located near the termination. 

Single-side supporting structures, structure-A and structure-C (Figure 1), were used in the following studies. Temperature dependence of the integrated intensity of structure-A and structure-C with the thickness of 25, 110, 230 and 760 nm was shown in Appendix A. T_g_s of structure-A with the thickness of 25, 110, 230 and 760 nm were 422.14, 419.76, 418.75 and 415.60 K respectively, whereas the T_g_s of structure-C were 418.39, 418.27, 417.97 and 416.05 K, as shown in Figure 3b. The error bars represented the average T_g_s by three groups of repetitive experiments. The obtained T_g_s indicated the interaction between the substrate-PMMA interface, corresponding to the previous studies [8,29,30]. It could be observed that the PMMA film supported on silver substrates exhibited lower T_g_ (0.45 to 3.75 K) than quartz substrates, suggesting that the PMMA chains on quartz substrates were considered as a result of the hydrogen bond between the PMMA and quartz. The strong interaction could lead to increased density near the interface with the increased T_g_. The interaction between the PMMA and silver could be considered as supramolecular ion-dipole interactions based on the oxygen atoms and defects on the surface of silver, which was generally weaker than the hydrogen bonds [31]. Thus the T_g_ of thin TC1-labeled PMMA film on the silver substrate was lower than the quartz substrate, and T_g_s for these two substrates tended to be consistent with the one of the bulk materials as the interfacial effects decreased with increased film thickness.

In Figure 3b, the T_g_ of structure-C declined slower than that of structure-A with the increasing thickness, which might be caused by the plasmonic effect of the silver substrate. The effects of thickness on T_g_ of the structure-C came from not only the interface effect, but also the plasmonic effect, while the effects on T_g_ from the structure-A came from only the interface effect. The plasmonic effect led to the amplified fluorescence signal on the interface effect, so that the fluorescence signal from the interface could be not ignored even with a thick TC1-labeled PMMA part far from the interface, leading to the slower decline of the T_g_ with the increased thickness in structure-C, as shown in Figure 3b.

### 3.3. Cause of the Interfacial Effect on T_g_ of PMMA Films

Raman spectroscopy was measured to monitor the molecule orientation on the interface, which was essential to the variation of T_g_ on different substrates. The unlabeled PMMA was chosen in the Raman measurement because of the strong fluorescence background of TC1 probe molecules with 532 nm excitation light, which might cover the Raman signals, based on the assumption that the presence of trace TC1 neither affected the orientation nor the conformation of PMMA macromolecules. Figure 4a exhibited the Raman shift of PMMA bulk film on quartz substrates and the bare quartz substrates. Table 1 showed the assignments of Raman shifts of the PMMA bulk film on quartz substrates [32,33].

It could be observed that Raman shifts of samples supported on silver substrates were different from the ones on quartz substrates. In our group’s previous study, the Raman measurement was performed to determine the molecular orientation and conformation between polymer thin films and metal substrate. The changes of Raman shifts showed that the orientation and conformation of the PMMA molecule chain were different from bulk molecules due to the interaction between PMMA and silver substrate [33]. Relative intensity of band was adopted to obtain qualitative information of molecular orientation. Surface selection rule was used to analyze the orientation of the adsorbed molecule [34], which was based on electromagnetic (EM) theory. The EM field around the metallic particles would be increased by the surface plasmons, which enhanced the Raman scattered intensity. The groups were orientated perpendicularly to the surface with the increasing Raman intensity, whereas paralleled groups led to decreased intensity [35]. 

Figure 4b showed Raman spectra of PMMA ultrathin film (25 nm) supported on quartz or silver substrates. Compared with the Raman spectra of the bulk PMMA, the bands of C=O symmetric stretching vibration at 1730 cm^−1^ and O–CH_3_ bending vibration at 1455 cm^−1^ shifted to 1731 cm^−1^ and 1456 cm^−1^, respectively, which was considered as a result of the hydrogen bonding between the PMMA and the hydroxyl surface of the quartz substrates (while the band at 2954 cm^−1^ assigned to the C-H stretch vibration of CH_2_ shifts to 2953 cm^−1^, eliminating the experimental error) [36,37,38]. On the contrary, the bands of C=O symmetric stretching vibration and O–CH_3_ bending vibration of the PMMA shift to 1724 and 1447 cm^−1^ for the silver substrate, respectively. In addition, there was a new band in samples supported on silver at 986 cm^−1^, which belonged to the stretching vibration of O–C. The relative intensity of the samples on silver substrate increased obviously as compared with the samples on quartz substrates, suggesting that the considerable motion of these groups orientated perpendicularly to the long axis of the molecules. 

We further investigated the Raman spectra of PMMA ultrathin film (25 nm) and bulk film (760 nm). As the thickness decreased, the proportion of molecules at the interface was gradually increasing in comparison to the molecules through the whole film, which led to more significant interfacial effects. Therefore, the orientation of groups at interface and in bulk were explored by the changes of the Raman intensity between ultrathin and bulk films. The Raman enhancement factor of PMMA films (25 and 760 nm) on silver substrates when compared with quartz substrates were shown in Figure 4c. The results indicated the enhancement factor at 2954 cm^−1^ of thin film was decreasing (α-CH_3_ orientated parallelly to the surface of silver substrates), while the enhancement factor at 1447 cm^−1^ corresponded to the groups of O–CH_3_ bending increased. The electromagnetic field on silver surface would increase the Raman signals of the groups vibrating in perpendicular directions, and decrease those in parallel directions [35]. Thus the groups of O–CH_3_ could be perpendicular to the surface, suggesting that the methyl ester groups matched with structure of trans-conformer adsorbed on the silver substrates as shown in Figure 4d. This result was consistent with another repetitive batch as shown in Appendix A.

Based on the discussion above, the Raman spectra supported the hydrogen bonding on the PMMA/quartz and the surface orientation via chemisorption on the PMMA/silver. Owing to the electronic coupling of unshared electrons pair in molecules and substrate surface in atomic-scale roughness, the Raman bands of some groups shifted (e.g., Ag–O), which led to the changes of molecular bonding geometries on silver surface [32]. We suggested that the adsorption of Ag-O=C might be the major interaction in structure-C. The interfacial orientation and conformation of molecules suggested that the exposing electronegative oxygen atoms with a pair of unshared electrons tended to the silver substrate, forming strong interactions between the ester groups and the metal surface [39]. Some researchers believed that the chelate-like complex might be formed in this circumstance [40]. There existed a certain interfacial strength at the interface of PMMA film and silver substrate, resulting in the increase of T_g_ of supported samples as compared with bulk powder and unsupported samples. Besides the absorption effect, hydrogen bonds (C=O⋯HX) existed between PMMA molecules and the quartz surface [41]. The hydrogen bonds were formed by two pairs of unshared electrons of two oxygen atoms in each methacrylate side group and hydroxy groups of silicon dioxide covering on quartz substrates under ambient conditions [42]. 

### 3.4. The Interfacial Interaction Energy between PMMA Molecules and Substrates by MD Simulation

In order to describe the variation of T_g_ of PMMA thin film on different substrates resulting from interfacial molecular conformation, the MD simulation was performed to explain interfacial interaction energy between PMMA molecules and substrates of silver and hydroxylated-quartz (Figure 5). We simulated the potential energy of the whole system containing all atoms, PMMA, silver, and quartz, which were summarized in Table 2. The interaction energies E_int_ were calculated by Equation (1), referring to the non-bond interaction energy. In this simulation, the vdW energy and electrostatic energy were the two main contributions to the non-bond interaction energy in this configuration. Also, the non-bond interaction energy, vdW energy and Electrostatic energy of each constituent parts could be extracted by the simulation results. For rigorous comparison of the relative interaction characteristics between PMMA and substrate in each configuration, the non-bond interaction energy between the PMMA and substrate were normalized by the contact area as shown in Table 2.

As shown in Table 2, the normalized interaction energy of PMMA/hydroxylated-quartz and PMMA/ silver were about −0.1191 and −0.0056 eV/Å^2^. A negative value indicated a decent binding between the PMMA molecules and substrate, as the lower the adhesion energy was, the stronger the interaction became. When compared with silver substrates, hydroxylated-quartz had strong adhesion energy to the PMMA molecule governed by electrostatic interaction, which was contributed to the hydrogen bonds. At the same time, the fluorescence measurement result further confirmed that the interaction in structure-A was stronger than structure-C basing on higher adhesion energy with hydrogen bonds on quartz substrates. As a result, T_g_s of PMMA film supported on quartz substrates exhibited higher value than the ones supported by silver substrates due to the lower mobility of the PMMA molecules.

### 3.5. Free Surface Effect on T_g_ of PMMA Films

To further understand the surface effect in molecule mobility behavior, we investigated the T_g_ of TC1-labeled PMMA thin film sandwiched between the substrate (quartz or silver) and PMMA bulk layer. From free surface to polymer/substrate interface, the polymer chain in thin film with different deepness could exhibit different molecule mobility [17,18,43]. Therefore, structure-B and structure-D (Figure 1) were prepared to study the free surface effects on the samples. 

As shown in Figure 6, the T_g_s of structure-B with film thickness of 25, 110, 230 and 760 nm were 423.74, 421.45, 418.74 and 415.39 K, respectively, whereas T_g_s of structure-D were 420.76, 420.01, 418.84 and 416.38 K. T_g_s of structure B and structure D increased (about 0.49 to 9.03 K respectively) when compared with those without coatings (structure A and structure C), which could attribute to the free surface effect. For polymer films, the T_g_ around the near-surface region might be different from the inner bulk region [44], which was due to the depressed entanglement density at the free surface of glassy polymer. As the molecular chain in the near-surface region had higher free volume and lower entanglement density, the mobility of molecules in the near-surface region was stronger than in the bulk region, resulting in the decrease of T_g_ [45,46]. Therefore, free surface could be suppressed by coating a bulk layer, which led to the increase of T_g_ in these structures.

## 4. Conclusions

In conclusion, the T_g_s of a series of TC1-labeled PMMA with varied film thickness were investigated by fluorescence spectroscopy. The results showed the increased T_g_s near the quartz substrate and silver substrate due to the interfacial interaction. The film on the silver substrate exhibited lower T_g_s than the one on quartz due to weaker interaction but the T_g_ decreased slower with the thickness as a result of the plasmonic fluorescence enhancement. Coating bulk PMMA layer on the films played a role in repairing free surface partially, leading to a higher T_g_ value. The MD simulation results and the Raman spectroscopy supported the hydrogen bonding on the PMMA/quartz and the surface orientation via chemisorption on the PMMA/silver, which impacted the cooperative segmental mobility associated with increased T_g_ due to the surface interactions. This work could have potential in the research on T_g_ and molecule conformation.

## Figures and Tables

**Figure 1 polymers-11-00601-f001:**
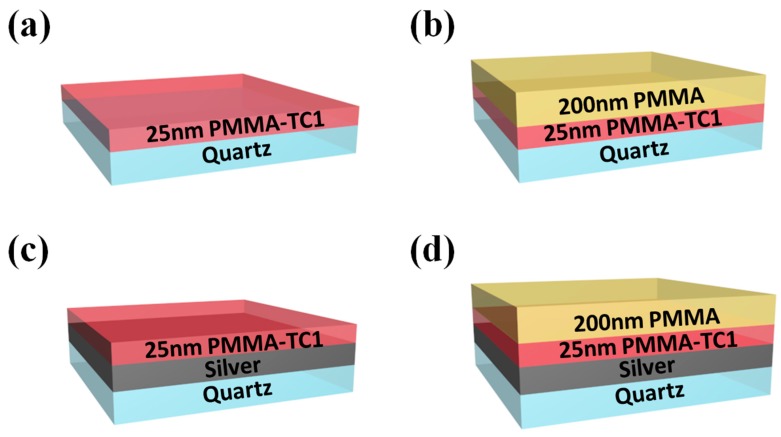
The schematic diagram of four structures of TC1-labeled PMMA denoting as (**a**) structure-A (quartz/PMMA-TC1), (**b**) structure-B (quartz/PMMA-TC1/bulk layer), (**c**) structure-C (quartz/Ag/PMMA-TC1), and (**d**) structure-D (quartz/Ag/PMMA-TC1/bulk layer).

**Figure 2 polymers-11-00601-f002:**
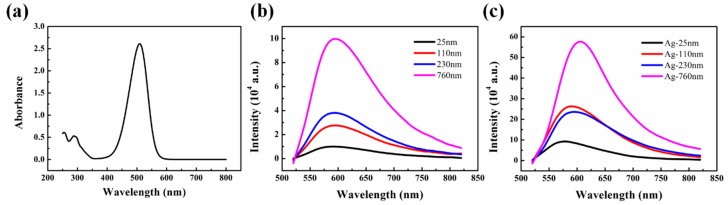
(**a**) UV absorption of TC1-labeled PMMA (*c* = 1.92 mg/mL in tetrahydrofuran). Fluorescence emission spectrum of TC1-labeled PMMA with thickness of 25, 110, 230 and 760 nm on (**b**) quartz and (**c**) silver substrate.

**Figure 3 polymers-11-00601-f003:**
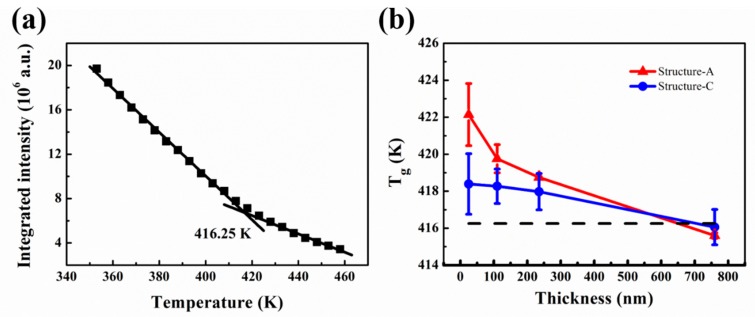
(**a**) Temperature dependence of the integrated intensity of TC1-labeled PMMA bulk film. (**b**) The spatial T_g_ of different thickness of TC1-labeled PMMA film for structure-A and structure-C: the T_g_s of each three films in the plot are defined by its average temperature. The dashed lined indicates the T_g_ of bulk TC1-labeled PMMA film supported on quartz substrate.

**Figure 4 polymers-11-00601-f004:**
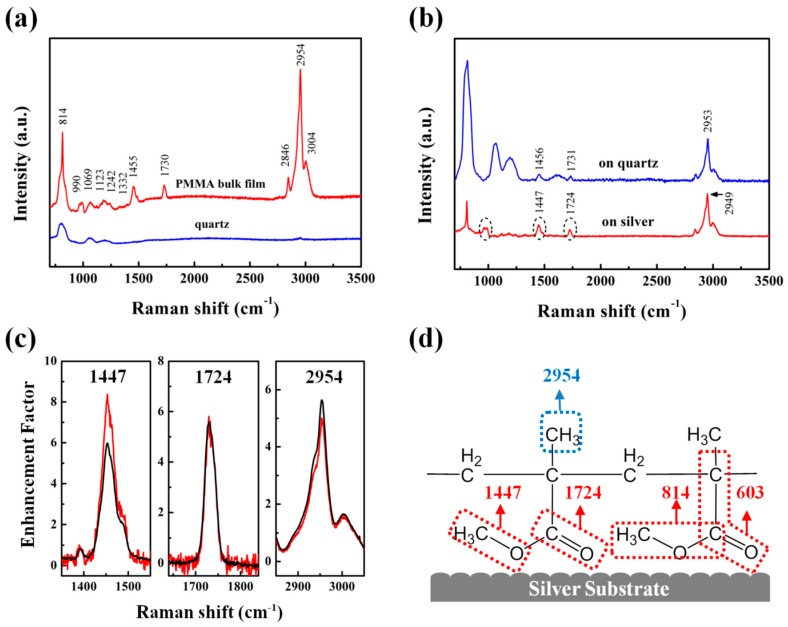
(**a**) Raman spectra of PMMA bulk film (760 nm) and bare quartz substrate. (**b**) Raman spectra (normalized with the intensity at 2949 cm^−1^ and 2953 cm^−1^) of PMMA ultrathin film (25 nm) supported on quartz (blue line) and silver (red line) substrates. (**c**) Raman enhancement factors of PMMA bulk film (760 nm, black line) and thin film (25 nm, red line) supported on silver substrates. (**d**) Possible structure of adsorbed ester group of PMMA molecule supported on silver substrates.

**Figure 5 polymers-11-00601-f005:**
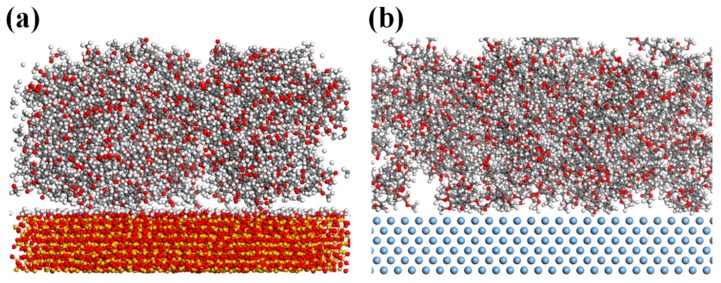
Snapshots of the simulated PMMA molecules adsorbed on (**a**) the hydroxylated-quartz surface and (**b**) the silver surface in x-z plane profile.

**Figure 6 polymers-11-00601-f006:**
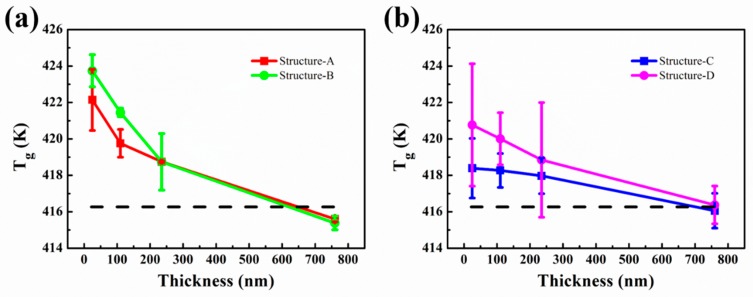
The spatial T_g_s of different thickness of TC1-labeled PMMA film for (**a**) structure-A/B and (**b**) structure-C/D: the T_g_s of each film in the plot are defined by its average temperature, and the black dash line indicates the T_g_ of the bulk TC1-labeled PMMA film supported on the quartz substrate.

**Table 1 polymers-11-00601-t001:** Assignments of Raman spectra of PMMA bulk film on quartz substrate.

Raman Shift [cm^−1^]	Mode	Assignment
3004	νa(O–CH_3_)	Asymmetric ^[31]^
2954	νa (α-CH_3_)	Asymmetric
2046	νs (CH_2_)	Symmetric
1730	νs (C=O)	Symmetric
1455	δs(O–CH_3_)	Bending ^[31]^
1332	τ or ω (CH_2_)	Twisting
1242	ν(C–O)	Stretching
1123	ν(C–C)	Stretching
990	ν(O–C)	Rocking
814	νs (C–O–C)	Symmetric

ν, stretching (s, symmetric; a, asymmetric); δ, bending; τ, twisting; ω, wagging.

**Table 2 polymers-11-00601-t002:** Simulative energy between PMMA and substrates of hydroxylated-quartz and silver.

System (Substrate)	E_total_	E_PMMA_	E_sub_	E_int_		Area [Å^2^]	Normalized Interaction Energy [eV/Å^2^]
				vdW Energy [eV]	Electrostatic Energy [eV]		
Hydroxylated-quartz	146.91	786.31	−2.63	−4.91	−631.84	5345	−0.1191
Silver	869.88	900.20	0	−30.32	0	5408	−0.0056

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
