# Peer review of "The Glass-Transition Temperature of Supported PMMA Thin Films with Hydrogen Bond/Plasmonic Interface"

_polymers, 2019, doi:10.3390/polym11040601_

Round 1

Reviewer 1 Report

The manuscript reports an in-depth caractherization on the effect of the interface on the glass transition temperature (Tg) of PMMA. This is carried out by fluorescence and Raman spectroscopy, in combination with molecular dynamics simulations. The main outcome of the study is that quartz substrate induces larger positive Tg deviations than silver substrate. This is basically due to the fact that in the former case hydrogen bonding is formed whereas the interaction PMMA/silver weaker Van der Waals bonding.

The results of the present study are interesting and deserve being presented in the long-standing debate of what causes Tg deviations in confinement. As such, publication of the manuscript is recommended after the following points are carefully addressed by the authors:

1.       The concept expressed in lines 212-216 is rather obscure. It is not clear whatsoever why “the plasmonic surface enhancement” should lead to “slower decline of the Tg with increased thickness”. In comparison to what? To PMMAon quartz maybe? If so, why? This part needs to be discussed in a more extensive and clear way.

2.       In lines 266-268, it is not clear the relation between enhancement of Raman band and the orientation of PMMA on silver substrate. A detailed and comprehensive explanation is warranted

3.       Is the MD simulations needed only to extract interaction energies? If yes, the authors must explain in details how this is operatively done. The explanation provided in the experimental section appears to be insufficient.

4.       The y axis in Figure 3b and 6 is the Tg not just a generic temperature.

5.       In line 186 the way Tg is defined must be provided. Is this the mid-point of the specific step in DSC?

6.       Concerning the effect of capping on the Tg of a film, a study reporting similar results as those of the present study must be quoted: Macromolecules, 2016, 49, 4647–4655.

Author Response

aaaa

Reviewer 2 Report

Jiayao Chen, et al. investigated the glass-transition temperature (Tg) of supported PMMA thin films with hydrogen bond/plasmonic interface. The authors investigated the Tg of PMMA films with different thickness and substrates using fluorescence measurement.  They found that the Tg increases with the decrease of film thickness. They also found the delayed decrease of Tg with the increase of the film thickness. They also investigated the hydrogen bonds of PMMA/quartz and molecular orientations of PMMA/silver by the Raman spectroscopy and the interfacial interaction energy by molecular dynamics simulation. Basically, this is a good study with both experimental and simulation investigations. I just have a suggestion for a minor revision. 

1.    If possible, please provide the photos of the four TC1-labeled PMMA in Fig. 1. 

2.    Please provide the statistic information for the curves in Fig. 2 and 4, such as the repeatability and how many times has been repeated. And, whether the curves shown here are the mean of the many repeated data. If not, please explain whether these data are representative of all data.

3.    Similarly, for Fig. 3 and 6, please explain the meaning of the error bars. 

Author Response

The respose to the comments has been uploaded as a Word file.
